# Adenovirus Vaccine Containing Truncated SARS-CoV-2 Spike Protein S1 Subunit Leads to a Specific Immune Response in Mice

**DOI:** 10.3390/vaccines11020429

**Published:** 2023-02-13

**Authors:** Keda Chen, Danrong Shi, Chaonan Li, Zhongbiao Fang, Yikai Guo, Wenjie Jiang, Jiaxuan Li, Hongyu Li, Hangping Yao

**Affiliations:** 1Shulan International Medical College, Zhejiang Shuren University, Hangzhou 310015, China; 2State Key Laboratory for Diagnosis and Treatment of Infectious Diseases, National Clinical Research Center for Infectious Diseases, Collaborative Innovation Center for Diagnosis and Treatment of Infectious Diseases, The First Affiliated Hospital, School of Medicine, Zhejiang University, Hangzhou 310000, China

**Keywords:** SARS-CoV-2, vaccines, adenoviral vector, S1 gene, immunity

## Abstract

The development of an efficient and safe coronavirus disease 2019 (COVID-19) vaccine is a crucial approach for managing the severe acute respiratory disease coronavirus 2 (SARS-CoV-2) pandemic in light of current conditions. In this study, we produced a shortened segment of the optimized SARS-CoV-2 spike gene (2043 bp, termed S1) that was able to encode a truncated S1 protein. The protein was tested to determine if it could elicit efficient immunization in mice against SARS-CoV-2. The presence of the S1 protein was confirmed with immunofluorescence and Western blotting. An adenovirus vaccine bearing the S1 gene fragment (Ad-S1) was administered intramuscularly to mice four times over 4 weeks. SARS-CoV-2 S1 protein humoral immunity was demonstrated in all immunized mice. The serum from immunized mice demonstrated excellent anti-infection activity in vitro. A robust humoral immune response against SARS-CoV-2 was observed in the mice after vaccination with Ad-S1, suggesting that the adenovirus vaccine may aid the development of vaccines against SARS-CoV-2 and other genetically distinct viruses.

## 1. Introduction

Coronavirus disease 2019 (COVID-19) is often associated with multiple organ failure and high mortality rates [1] and poses a major threat to public safety worldwide. As of 15 July 2022, the COVID-19 pandemic has persisted globally, with 557,917,904 confirmed cases, including 6,358,899 deaths, reported to the World Health Organization (WHO). Furthermore, a total of 12,130,881,147 vaccine doses have been administered as of 11 July 2022. Therefore, designing and developing novel coronavirus (nCoV) vaccines are of great importance and one of the top global health priorities. The S protein is a key factor in the entry of the SARS-CoV-2 severe acute respiratory disease coronavirus 2 (SARS-CoV-2) into host cells [2,3]. The protein binds to the host receptor and allows the virus to enter the host cell smoothly [4]. N-linked glycans protrude from the trimer surface and decorate the S protein extensively, affecting S protein folding, humoral immunity, and host cell protease processing [5]. The SARS-CoV-2 S protein has lower N-linked glycan density than other human pathogenic coronavirus-induced S proteins [6]. Therefore, the S protein may be highly immunogenic and is a major target of neutralizing antibodies. The S protein has three structural domains, among which the S1 domain is the most important S protein surface antigen [4]. Due to the difficulty of producing large recombinant proteins (the extracellular domain of the protein S is about 1300 amino acids) and the risk of antibody-dependent enhancement (ADE) of infection, S1 (about 700 amino acids) and its receptor-binding domain (RBD, about 200 amino acids) are widely regarded as the most attractive potential targets for a coronavirus vaccine [7,8]. Human adenovirus serotype 5(HAdV-C5) is widely used in basic virology as a gene therapy and vaccine delivery vector [9]. HAdV-C5 has natural diversity and can parasitize most hosts, which forms the basis for the development of animal experiments. Recombinant adenovirus vectors with replicability and replication defects constructed from HAdV-C5 have been widely used in vaccine delivery and gene therapy [9,10]. Currently, adenoviruses have been approved as vaccines against acute respiratory infections and there have been many trials of such vaccines, such as for malaria and HIV-1 [9]. In this paper, we describe a SARS-CoV-2 human replication-deficient adenovirus vector vaccine and its preparation and application.

In this study, the adenovirus vaccine was constructed as follows: the vector was human replication-defective adenovirus HAdV-C5 with E1 and E3 deletion, the skeleton plasmid was recombinant adenovirus with pBHGloxΔE1,3Cre, the packaging cell line was HEK293 cells, and the target gene was the truncated S1 protein gene of SARS-CoV-2. S1 protein expression in the vaccine was identified by Western blot and immunofluorescence assays. To determine the quality of immune responses induced by the spike-based vaccine candidates, the post-vaccination antibody response was examined in detail. The immune benefit of the vaccine was evaluated with a binding antibody test (enzyme-linked immunosorbent assay [ELISA]) and neutralization antibody test. Our findings provide the basis for the development and evaluation of COVID-19 vaccines and treatments based on the S1 protein.

## 2. Materials and Method

### 2.1. Cells and Animals

HEK293 and Vero E6 monkey kidney cell lines were used in this study. Both cell lines were grown in Dulbecco’s modified Eagle’s medium (DMEM) supplemented with 2% fetal bovine serum (FBS) and 1% penicillin and streptomycin at 37 °C with 5% CO_2_ and saturated humidity. 

Twenty female C57BL/6 mice (6–8 weeks) were purchased from the Zhejiang Animal Experimental Center, Zhejiang Province. The mice were randomly divided into two groups with 10 mice in each group. One group was immunized (100 μL intraperitoneal injection) with an Ad5 vector expressing the SARS-CoV-2 spike protein (Ad5-S) and the other with PBS (100 μL intraperitoneal injection) as a control. The total amount of virus was 5 × 10^9^ V.P. All mice were immunized with D0/D14 intramuscular injection [11,12]. At D14, peripheral blood of 100 μL was taken, serum was separated, and the second injection was given two hours after blood collection. At D28, blood was collected by retro-orbital puncture and serum was separated. Binding antibodies were detected 3 days after each blood collection, neutralizing antibodies were detected 7 days after each blood collection, and cytokines were detected 7 days after the second blood collection. All mice were euthanized; all tests were conducted in strict accordance with the “Guidelines for The Care and Use of Experimental Animals of the People’s Republic of China” and approved by the Zhejiang Shuren University Ethical Council.

### 2.2. Vaccines

All the SARS-CoV-2 strains used in this study were collected from COVID-19 patients with consent. The state Key Laboratory for Diagnosis and Treatment of Infectious Diseases developed an adenovirus vaccine (Vero cell, WuHan strain, GISAID number: EPI_ISL_415711) against SARS-CoV-2 and then the vaccine was manufactured in a biosafety level (BSL)-compliant setting. The vaccine specification is 0.5 mL/dose and contains Tris, NaCl, cane sugar, MgCl2.6H2O, C6H9N3O2, absolute ethanol and the SARS-CoV-2 S1 protein.

### 2.3. Recombinant Adenovirus Construction

The SARS-CoV-2 S1 protein sequence (no. QRU91950.1) was obtained from PubMed. According to the gene degeneracy, the optimized codon nucleotide sequence suitable for mammalian cell expression was designed based on the premise that the amino acid sequence of the product protein encoded by the SARS-CoV-2 S1 protein remained unchanged. The total length was 2043 bp. The 5′ precursor sequence was synthesized with tissue plasminogen activator (tPA) and the sequence encoding SARS-CoV-2 S1 amino acid 2–688 and the tPA precursor sequence were ligated. A Kozak sequence and *Spe*I restriction site were added in front of the start codon and the *Xba*I restriction site was added after the termination codon. The nucleotide sequences of the above genes were synthesized and cloned into pUC18 by Sangon Biotech to obtain the cloned plasmid of the synthetic gene. The synthesized S1 gene sequence and vector pDC316 were digested with restriction endonucleases *Spe*I and *Xba*I. The target fragment and vector were retrieved from the gel and connected to form a shuttle plasmid. The SARS-CoV-2 shuttle plasmid was packaged with the AdMax adenovirus system skeleton plasmid pBHGloxdΔE1 and 3Cre by co-transfection of HEK293 cells. The primary virus solution was obtained after repeated freeze-thawing and centrifugation, and the virus was amplified and cultured after plaque purification.

### 2.4. Determination of Adenovirus Titer

HEK293 cells in good health were selected and resuspended to prepare a 5.0 × 10^5^ cells/mL cell suspension, which was seeded in a 24-well plate (1 mL/well) and cultured at 37 °C in a 5% CO_2_ environment. The sample was serially diluted tenfold and a diluted sample (0.1 mL) of 10^−5^ to 10^−8^ cells was inoculated onto a 24-well plate. Then, the plates were exposed to infection at 37 °C with 5% CO_2_ for 48 h. Subsequently, the culture medium was removed, pre-cooled methanol (0.5 mL) was added, and the samples were fixed at −20 °C for 20 min. After fixation, the samples were washed with phosphate-buffered saline (PBS) and blocked with 1% bovine serum albumin (BSA, 0.2 mL) at 37 °C for 1 h. Next, the primary (Mouse Anti-Adenovirus Hexon, AbD Serotec 04000079, 1:2000) and secondary (Goat pAb to MS IgG2a(HRP), Abcam ab97245, 1:1000) antibodies were added and incubated for 1 h, during which PBS was used for washing. After incubation, newly prepared working solution (0.2 mL) was added and incubated for 5–10 min at room temperature. Then, the working solution was discarded, the samples were washed with PBS, and PBS (1 mL) was added again. The average number of positive cells in the microscopic field (cat. CKX53, OLYMPUS Research Inverted System Microscope) was calculated, where a gradient with 5–50 positive cells in the field of view was selected and at least five areas were randomly selected for counting. The number of visual fields in each well of the 24-well plate was calculated. Then, the virus titer was calculated according to the following Formula (1):(1)$$Viral\;Titer\;\left(\mathrm{pfu}/\mathrm{mL}\right)=\frac{\left(average\;positive\;cells/field\right)\times \left(79\;fields/well\right)\times \left(dilution\;factor\right)}{\left(0.1\;\mathrm{mL}\right)}$$

### 2.5. Real-Time PCR

To detect HEK293 cell mRNA expression following viral infection, we extracted intermediate RNA according to the TRIzol operating instructions (Invitrogen, Waltham, MA, USA). Total RNA was extracted using TRIzol and processed with RQ1 RNase-free DNase I (Promega, Madison, WI, USA) to eliminate residual DNA. Complementary DNA (cDNA) was obtained by reverse transcription of the extracted RNA using a PrimerScript™ RT Reagent Kit with gDNA Eraser Kit (Takara, Kusatsu, Japan). DNA fragments were generated with the specific primers P1 (5′-GGTGATTCTTCTTCAGGTTGGA-3′) and P2 (5′-GTTTCTGAGAGAGGGTCAAGTG-3′) of the target gene (S1). The gene was amplified with primers P3 (5′-GTCTTCACCACCATGGAGAA-3′) and P4 (5′-TAAGCAGTTGGTGGTGCAG-3′) as the internal control (GAPHD).

### 2.6. Western Blot

The cell lysate and supernatant were separated with sodium dodecyl sulfate–polyacrylamide gel electrophoresis (SDS-PAGE), then transferred to polyvinylidene fluoride (PVDF) membranes. The blots were visualized with Western blot imaging equipment (Bio-Rad, Hercules, CA, USA). Briefly, the membranes were transferred to TBST (50 mL Tris-HCl, pH 7.5, 8 g NaCl [0.5 mL], 0.2 g KCl, 0.5 mL Tween-20) with 5% non-fat dry milk powder and shaken on a decolorizing shaker at room temperature for 1 h. The membranes of the experimental and control groups were removed from the sealing solution and incubated with the primary antibody: [SARS-CoV-2 (2019-nCoV) Spike Antibody (1:5000, cat. #40591-T62, Sino Biological, Beijing, China)] at room temperature. The primary antibodies were shaken and incubated overnight on a decolorizing shaker at 4 °C. The blot was rinsed with TBST solution and incubated for 2 h with the secondary antibody [goat anti-rabbit immunoglobulin G (IgG) H&L (HRP) (1:10,000, cat. #ab6721, abcam, Cambridge, UK)]. After rinsing, the blots were incubated for 1 min with a LumiBest ECL Substrate solution kit (cat. #4AW011-100, 4A Biotech), then observed in an imager.

### 2.7. Immunofluorescence Assay

HEK293 cells (3 × 10^6^/well) were seeded into 12-well plates. After 48 h, the cells were infected with adenovirus containing the S1 gene (Ad-S1). After 48 h, the medium was aspirated, the cells were washed once with PBS containing 2% BSA, and fixed for 15 min with methanol that had been pre-chilled for 15 min at −20 °C. After three washes with 2% BSA in PBS, the cells were permeabilized with 0.5% Triton X-100 for 10 min. Then, the cells were blocked for 1 h with 2 mL 2% BSA, the blocking solution discarded, and the cells were washed three times with PBS. Subsequently, 2 mL primary antibody [SARS-CoV-2 (2019-nCoV) Spike Antibody (1:2000, cat. #40591-T62, Sino Biological)] was added to each well and incubated at room temperature for 1 h or 4 °C overnight. Then, the sample was rinsed three times with 2% BSA for 5 min per rinse. Next, 2 mL secondary antibody [goat anti-rabbit IgG H&L (Alexa Fluor 488) (1:1000, cat. #550037, ZenBio)] was added to each well and incubated at room temperature for 1 h. Subsequently, the samples were rinsed three times with 2% BSA for 5 min per rinse. 4′,6-diamidino-2-phenylindole (DAPI) solution (2 mL, 1 mg/mL) (1:400, cat. #S0001, Bioss, Woburn, MA, USA) was added to each well and reacted for 10 min in the dark. The analysis was observed with an EVOS™ M7000 imaging system (cat. #AMF7000, Invitrogen).

### 2.8. ELISA

SARS-CoV-2 S protein (1 μg/mL) was coated overnight in 96-well plates (100 μL/well) with 0.05 M bicarbonate buffer. After washing with PBST (0.2 g KH2PO4, 2.9 g Na2HPO4·12H_2_O, 8.0 g NaCl, 0.2 g KCl, 0.5 mL Tween-20, add water to 1000 mL) five times, blocking solution was added and incubated at 37 °C for 1 h. The serum sample was diluted and added to each well (100 μL/well). After 1 h incubation at 37 °C, the samples were washed with PBST five times. The primary antibody [SARS-CoV-2 (2019-nCoV) Spike Antibody (1:2000, cat. #40591-T62, Sino Biological)] was added and incubated for 1 h, then the samples were washed five times with PBST, followed by 1 h incubation at 37 °C with horseradish peroxidase-conjugated secondary antibody [goat anti-rabbit IgG H&L (HRP) (1:10,000, cat. #ab6721, abcam)] and five washes with PBST. After adding 3, 30, 5, 50-tetramethylbiphenyl anhydride for 5 min, the reaction was stopped with 2 M sulfuric acid. The optical density was measured at 450 nm and 630 nm by an enzyme-labeling instrument (Molecular Devices, SpectraMax 190) and then fitted to a standard curve.

### 2.9. Cytokine Determination

The plate (cat. #T-k15048D-1, MSD) was washed three times with 150 μL/well wash buffer and 50 μL prepared sample was added to each well. Then, the plate was sealed with an adhesive plate seal and incubated at room temperature with shaking for 2 h. Next, the plate was washed three times with 150 μL/well wash buffer and 25 μL detection antibody solution was added to each well. The plate was sealed again and incubated at room temperature with shaking for 2 h. Next, the plate was washed three times with at least 150 μL/well wash buffer; 150 μL 2× Read Buffer T (MSD) was added to each well, and the plate was analyzed on an MSD instrument.

### 2.10. Neutralizing Antibody

#### 2.10.1. Pseudovirus Neutralization Assay

The mouse serum neutralization activity was tested using a vesicular stomatitis virus pseudoviral system (VSV). Serum was inactivated in a water bath for 0.5 h at 56 °C, and then serially diluted to the required range. HEK293 cells were plated in a 96-well plate. The diluted serum was mixed with 200 CCID50 (the virus dose that can infect 50% of the cell culture)/100 μL of the virus suspension in a ratio of 1:1 and placed in a CO_2_ incubator (37 ± 1 °C) for 2 h. Next, 100 μL of virus maintenance solution containing 0.5% penicillin-streptomycin double antibody was added to each well, and the neutralized mixture was inoculated into a 96-well plate containing cells at 100 μL per well, with two repeat wells for each dilution. At the same time, a normal cell control was set, and the cells were incubated in a CO_2_ incubator (37 ± 1 °C) for 96 h. Then, 200 CCID50/100 μL of virus solution was diluted to 10-1~10-3. The cell culture medium in the 96-well plate was discarded, 150 μL of the virus maintenance solution was added to each well, and the virus solution dilution was inoculated at a concentration of 10-1~10-3 into wells of the 96-well plate (50 μL per well). Each dilution was repeated in 8 wells; a normal cell control was set at the same time, followed by culture for 96 h. The changes in cell CPE were observed under an inverted microscope. The normal cell control should have no cytopathic changes, and the positive control should have CPE changes. Neutralization endpoints (serum dilutions converted to logarithms) were calculated by observing the CPE. The seronegative/positive criterion was 1:12 as the positive/negative criterion, <1:12 was negative, and ≥1:12 was positive.

#### 2.10.2. Live Neutralization Assay

The mouse serum neutralization activity was evaluated by a live neutralization test. The diluted serum was mixed with SARS-CoV-2 and incubated at 37 °C for 1 h. The mixture was added to a 96-well plate to infect the Vero E6 cells. After 72-h incubation at 37 °C, the virus cytopathic effect (CPE) was observed under ×40 magnification and the neutralization titer (the reciprocal of 50% serum dilution required to neutralize virus infection) was calculated. The above procedures were all carried out in a biosafety level 3 environment.

### 2.11. Statistical Analysis

Statistical analyses were performed by a two-sample *t*-test using SPSS 26. *p*-values ≤ 0.05 were considered significant. The central tendency was measured with the geometric mean titer (GMT).

## 3. Results

### 3.1. Recombinant Adenovirus Construction

The recombinant adenovirus shuttle plasmid pAdeno-CMV-S1 with correct insertion was obtained; its structure diagram is shown in Figure 1. The recombinant adenovirus contained the HAdV-C5 genome lacking the E1/E3 region. A schematic diagram of the recombinant adenovirus vector obtained by the shuttle plasmid and cytoskeleton plasmid transfection in HEK293 cells is shown in Figure 1.

### 3.2. Characterization of Recombinant Adenovirus

The recombinant plasmid containing the target gene was identified by PCR (see Figure 2A for gel electrophoresis results). This was a repeated experiment. The primer sequences were MCMV-F ggtataagaggcgcgaccag and S1-R acaataagtagggactgggtc, and sequencing confirmed that the sequence was consistent with the design. SARS-CoV-2 S1 expression in the cells was detected with Western blotting (Figure 2B) and indirect immunofluorescence (Figure 2C). The band coloration (~75 kDa) was detected by Western blotting and was consistent with the expected band position. At the transcriptional level, the in vitro expression of S1 was detected by PT-PCR. The results showed that the expression of S1 mRNA in HEK-293 cells was significantly higher than that in the control group.

### 3.3. Adenovirus Titer

In this experiment, an average of six positive cells were calculated in five microscopic fields, and the virus in the well was diluted 10^8^ times. Based on the formula described in the Materials and Method section, the adenovirus titer was 4.74 × 10^11^ plaque-forming units (pfu)/mL (Figure 2D).

### 3.4. Post-Vaccination Cellular Immunity

During the test period, there were statistically significant changes in serum cytokines (*p* ≤ 0.05), which mainly showed that the concentration of Keratinocyte chemoattractant/human growth-regulated oncogene (KC/GRO) was decreased, and the concentrations of IFN-γ, IL-10, IL-12p70, IL-1β, IL-2, IL-4, IL-5 and TNF-α were increased, while the concentration of IL-6 was not significantly changed (Figure 3).

### 3.5. Detection of Anti-SARS-CoV-2 S1 Antibodies Post-Immunization

ELISA revealed that SARS-CoV-2-specific antibodies were present in the mice injected with the first and second Ad-S1, but not those injected with the control adenovirus capsid or PBS solution (Ad/PBS) (Figure 4A). IgG titers of 1:2^10^ and 1:2^12^ were detected in the Ad-S1-vaccinated mice two weeks after the first and second immunizations, respectively. The dilution titer at 28 days post-vaccination (D28, GMT 2297.4) was 6.1-fold higher than that at D14 (GMT 378.9).

### 3.6. Neutralizing Antibody Assay

SARS-CoV-2 challenge of Vero E6 cells enabled the detection of neutralizing activity in the immunized mouse serum (Figure 4B,C). The D14 and D28 neutralization titers of the Ad-S1-immunized Vero E6 cells against SARS-CoV-2 live virus challenge in vitro were 1:2^4.3^ and 1:2^6.3^, respectively. The pseudovirus assay yielded similar results to those of the live virus assay, with neutralization titers of up to 1:2^8.1^ and 1:2^9.6^ at D14 and D28, respectively. The D28 pseudovirus neutralizing titer (GMT 769.0) was 2.7-fold higher than that of D14 (GMT 283.7).

## 4. Discussion

We have been working to develop vaccines to stop the COVID-19 pandemic and its rapid spread. The hunt for an effective vaccination against SARS-CoV-2 continues and existing vaccine development strategies include whole virus particle vaccines, live attenuated virus vaccines, purified virus subunit vaccines, and genetic vaccines [13,14,15,16,17]. The epitopes on spike proteins detected in infected serum are highly antigenic, with the ability to induce a strong humoral immune response and neutralizing antibodies in individuals infected with SARS-CoV-2 and recovered from COVID-19 [18,19]. The S protein appears to be an ideal target for vaccination against SARS-CoV-2 infection [20]. The S protein S1 subunit induced effective immunity against SARS-CoV-2 and protected against SARS-CoV-2 in animal models [21,22].

The SARS-CoV-2 truncated N protein has a better expression effect than N protein [23]. Based on this, we selected the truncated S1 protein in this experiment. In other studies, more than 90% of the neutralization activity of the Moderna mRNA vaccine was caused by the RBD antibody, and the neutralization titer directly decreased from >1000 to <25 after the RBD antibody had been eliminated [24]. Higher levels of RBD-specific IgG were associated with increased serum neutralization [25]. Evidently, the S protein demonstrates a greater immune response to the RBD region, and the S1 protein containing RBD can induce neutralizing antibodies with a higher titer than the S protein [26]. Antigens delivered by adenovirus vectors induce strong cellular and humoral immunity after a single immunization, making them useful as an emergency prevention tool in a pandemic [27]. Adenovirus-based vaccine strategies are therefore an important part of the fight against SARS-CoV-2 infection.

In this study, we constructed a recombinant adenovirus SARS-CoV-2 vaccine containing the SARS-CoV-2 S1 subunit and designated it Ad-S1. Like CanSinoBIO, which has been approved for marketing [28], we used HAdV-C5 in this vaccine. Adenovirus is widely used in recombinant gene therapy and as a vaccine vector. However, the seropositive rate of HAdV-C5 is as high as 75–80% in the normal population [29]. This implies that pre-storage immunity against HAdV-C5 vectors significantly reduces vaccine-induced immunity. Developed at the University of Oxford in the United Kingdom, ChAdOx1 nCoV-19 uses a chimpanzee adenovirus vector that, by contrast, has a lower seropositive rate in humans, and when positive, usually presents a lower serum antibody titer [30]. Therefore, adenovirus vectors that can evade pre-existing immunity can be explored to enhance the protective efficacy of the adenovirus vaccine.

In the present study, the serum status of mice before and after immunization was determined by ELISA and neutralization tests and yielded evidence of humoral immunity for vaccine candidates. The serum produced after intramuscular injection of the vaccine in mice effectively protected Vero E6 cells from SARS-CoV-2 infection in vitro (Figure 4). We also detected a stronger immune response and better protection in the mice after a second immunization (Figure 4). Our experimental results are similar to mainstream experimental results.

As SARS-CoV-2 research continues, several studies have demonstrated that the cytokine storm secondary to SARS-CoV-2 infection may affect T-cell production, making it difficult for patients to maintain long-term immunity to SARS-CoV-2 [31]. Therefore, it is necessary to determine whether Ad-S1 can induce T-cell-mediated immunity in mice. Our results demonstrated that IFN-γ and IL-12 concentrations increased, and IL-4 concentrations decreased in the Ad-S1 group, indicating that the vaccine induced Th1 cell survival and growth in the mice. IL-12, IL-10, and TNF-α secretion by the Th1 cells increased slightly in the experimental group. These results indicated that Ad-S1 effectively induced a Th1-biased T-cell response in the mice [32]. The increased IL-5 indicated that Th2 cells also participate in the immune response and produce certain effects. The KC/GRO concentration in the experimental group was significantly reduced compared to that of the PBS control group, which might have been due to the reduced neutrophil chemotaxis that suppressed the inflammatory response and allowed the vaccine side-effects to be controlled slightly. This would be more beneficial to immunocompromised people, such as the elderly or patients undergoing chemotherapy.

Li M., et al. [29] developed an adenovirus vaccine using Ad68 as a vector. The results showed that in vivo neutralizing activity of SARS-CoV-2 could be detected within two weeks after intramuscular immunization of BALB/c mice, and then continued to increase, and the neutralization antibody titer (PRNT ID50) reached 957.3 at eight weeks. Meanwhile, in the study of Liu J., et al. [33], vaccines using AdC6 and AdC68 as vectors were developed; both binding and neutralizing antibody responses after immunization were generated 14 days after dose, with an IgG titer of 7108 (geometric mean titer, GMT; AdC6) and 5489 (GMT, AdC68). The neutralization titer 50 (NT50) of the pseudovirus neutralization assay was 125 (GMT, AdC6) and 67 (GMT, AdC68). 

In the present study, mice vaccinated with Ad-S1 had IgG titers of 1:10^10^ and 1:12^2^ detected by ELISA two weeks after the first and second immunization, respectively. The dilution titer of ad-s1 was 378.9 (GMT) at 14 days after inoculation and 2297.4 (GMT) at 28 days after inoculation. In the neutralization antibody test, Vero E6 cells immunized with Ad-S1 showed 1:2^4.3^ and 1:2^6.3^ neutralization at D14 and D28 against live SARS-CoV-2 virus challenge in vitro, respectively. The neutralization titers of the pseudovirus at D14 and D28 were 283.7 (GMT) and 769.0 (GMT), respectively. Thus, the vaccine in this study was more effective in terms of the immune response in the mouse model.

Due to the limitation of the experimental conditions, we did not perform enzyme-linked immunosorbent spot (ELISpot) analysis of the spleen immune cells of the experimental mice. Moreover, attack protection experiments of SARS-CoV-2 on mice could not be performed due to the limitation of the physical protection laboratory. However, we determined that the vaccine had high efficacy in mouse models and could be used as an ideal reserve vaccine strain. In addition, the immunogenicity, safety, and effectiveness of prospective SARS-CoV-2 vaccine candidates should be examined in additional animal models (e.g., rabbits, primates, raccoons, dogs), as mouse models may not fully duplicate human immunological features.

## 5. Conclusions

The data obtained from our preclinical studies of the adenovirus vaccine demonstrated that the vaccine has sufficient safety and effectiveness. Therefore, it may be a promising and feasible prophylactic vaccine against SARS-CoV-2 infection.

## Figures and Tables

**Figure 1 vaccines-11-00429-f001:**
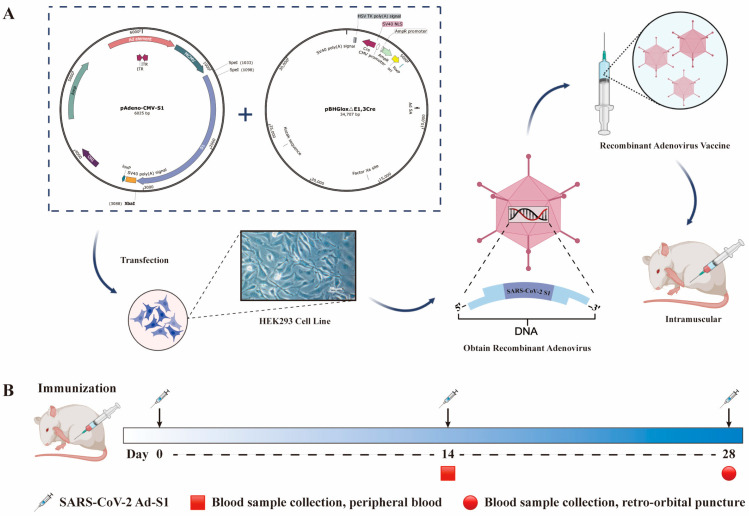
Construction of recombinant adenovirus vaccine and experimental strategy. (**A**) pAdeno-CMV-S1 is a recombinant adenovirus shuttle plasmid containing the target gene S1. pBHGloxΔE1,3Cre is the adenovirus skeleton plasmid. pBHGloxΔE1,3Cre and pAdeno-CMV-S1 were extracted with a QIAGEN plasmid extraction kit (Beijing North Yitao Trading Co., LTD., Beijing, China). One day before transfection, HEK293 cells were passaged in a 25 cm^2^ cell culture flask. The culture medium was DMEM containing 5% FBS without antibiotics. On the second day, 60–80% of the cells were selected and the transfection solution was added to the cells. After 7 days of transfection, the cells were scraped off, centrifuged, the supernatant discarded, and the cells were resuscitated with PBS. The cells were freeze-thawed repeatedly at −80 °C and 37 °C. After centrifugation, the supernatant contained the primary virus solution. The recombinant adenovirus strain was purified, amplified, and obtained after further processing and designated Ad-S1. The vaccine was injected intramuscularly into the mice for follow-up studies. (**B**) The depicted time scale of immunization and blood withdrawal.

**Figure 2 vaccines-11-00429-f002:**
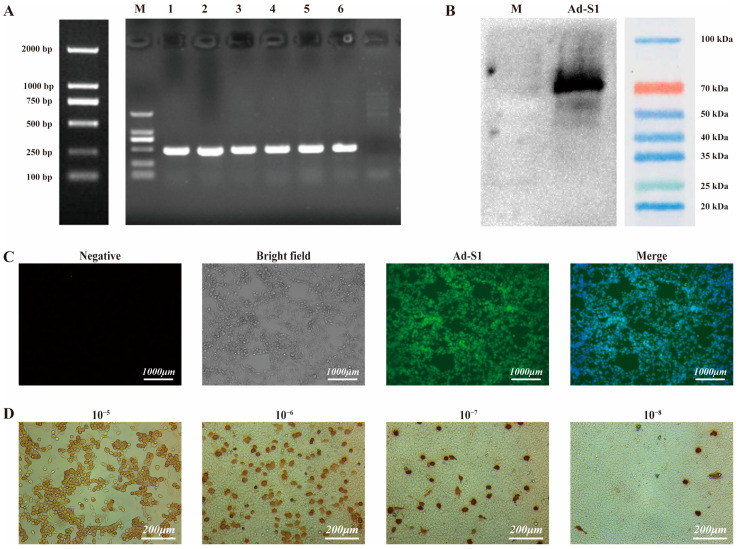
Identification of recombinant HAdV-C5–SARS-CoV-2 S1 protein and determination of adenovirus titer. (**A**) PCR detection of the recombinant plasmid with the target gene S1. (**B**) Western blot detection of SARS-CoV-2. Exponential-growth HEK293 cells were infected with SARS-CoV-2 for 48 h, then the cellular protein was extracted and separated by SDS-PAGE. SARS-CoV-2 expression was confirmed by Western blotting with goat anti-rabbit IgG. (**C**) Immunofluorescence microscopy of HEK-293 cells infected with Ad-S1 (green). Cells were counterstained with 4′, 6-diamidino-2-phenylindole (DAPI) to stain the nuclei. Scale bar 1000 μm. (**D**) Plaque assay was used for measurement. Micrographs depicting 10-5, 10-6, 10-7 and 10-8 dilution (from left to right).

**Figure 3 vaccines-11-00429-f003:**
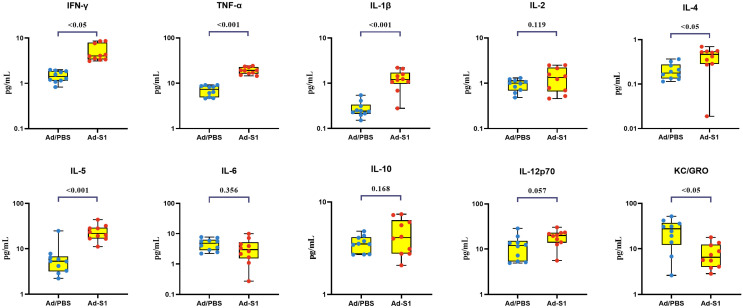
Ad-S1 induced altered plasma levels of type 1/2 cytokines, type 1 interferons, and other cytokines. The results were detected by drawing blood from the mice 7 days after the first immunization. The data are presented as box and whisker scatterplots. Each circle represents a single individual. *p*-values were calculated using a two-sample *t*-test.

**Figure 4 vaccines-11-00429-f004:**
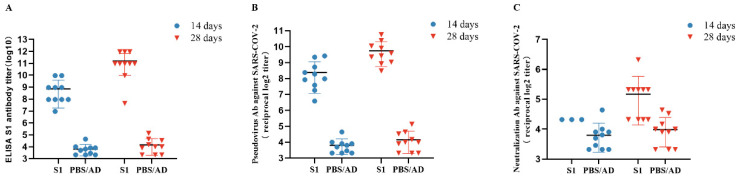
Ad-S1 induced high titers of antibodies (**A**) and neutralization activity (**B**,**C**) in mice. (**A**) ELISA of SARS-CoV-2-specific serum IgG from Ad-S1-immunized mice. (**B**) Analysis of pseudovirus activities of serum from the Ad-S1-immunized mice. (**C**) Analysis of neutralizing activities of serum from the Ad-S1-immunized mice.

## Data Availability

The raw data supporting the conclusions of this article will be made available by the authors, without undue reservation.

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
