# Peer review of "Adenovirus Vaccine Containing Truncated SARS-CoV-2 Spike Protein S1 Subunit Leads to a Specific Immune Response in Mice"

_vaccines, 2023, doi:10.3390/vaccines11020429_

Round 1
Reviewer 1 Report
The current manuscript entitled “Adenovirus vaccine containing truncated SARS-CoV-2 spike protein S1 subunit leads to a specific immune response in mice “ by Chen et al addresses an interesting study that illustrates that prominent immune response against SARS-CoV-2 was observed in the mice after vaccination with Adeno-S1. Moreover, the present study illustrated that adenovirus may target the development of vaccine against SARS-CoV-2 and other genetically diverse viruses.
However, the author needs to focus and address the following concerns in the present study.
1. The Author need to provide source of information which gathered regarding the vaccines
2. In the present study Mice has been used for Adeno viral infection/Injection source and protocol was missing in the Methodology section.
3. Author need to provide information regarding the blood withdraw from the mice.
4. Author need to explain the basis of administration of vaccine for four times over 4 weeks in the current study.
5. At first Short signs should be elaborate and utilize the entire Manuscript Draft. Example nCoV and at 33 line the same error was observed.
6. Please check 47 line, it has sentence error
7. Further Sampling /Number of mice has been used in the present study should be mentioned and Why sampling was different from figure 3 to Figure 4.
8. After how many days after serum was collected and tested for further follow up studies should mention in Methodology.
9. For the betterment of this manuscript author should include the depicted time scale of immunization and blood withdraw should be included.
10. In figure 2 Scale bar or Magnification of image was missing.
11. In the Methods the source of Cell lines and Passage should be clarified. Author has plated 5 *10 5 cells per ml but at the same time in the methods mention author mentioned good condition. Could you please elaborate good condition?
12. Author should describe after how many hours inoculate was treated should be explained in the methodology
13. In the methodology adeno virus titer determination, author should mention the microscopic field chosen the calculation.
14. In results section Figure 3 should categorized into A-J for simplifying viewer’s View.
15. The author need to recheck check IL beta and /KC/GRO calculations. Since the down ward is error bar is too high is it falls under 0.005 and 0.023 statistics?
16. In the figure legends the author needs to mention how many animals were used per group should be mentioned. In the Figure 4 C why only three animals were used?
17. In the line 290, Reference need to include for his statement.
18. In the conclusion the author stated the adenovirus vaccine demonstrated that the vaccine has enough safety and effectiveness, but we did not see any type of the safety data in mice through out the study. Please address the above statement.
19. Why the first four reference style is different from the rest?
Author Response
Response to Reviewer 1 Comments
The current manuscript entitled “Adenovirus vaccine containing truncated SARS-CoV-2 spike protein S1 subunit leads to a specific immune response in mice “ by Chen et al addresses an interesting study that illustrates that prominent immune response against SARS-CoV-2 was observed in the mice after vaccination with Adeno-S1. Moreover, the present study illustrated that adenovirus may target the development of vaccine against SARS-CoV-2 and other genetically diverse viruses.
However, the author needs to focus and address the following concerns in the present study.
Point 1: The Author need to provide source of information which gathered regarding the vaccines
Response 1: Thank you very much. The comments and suggestions made by you have helped us improve the manuscript significantly. Information about vaccines has been added in Methodology 2.2. The change has been highlighted in yellow in the revised manuscript.
Point 2: In the present study Mice has been used for Adeno viral infection/Injection source and protocol was missing in the Methodology section.
Response 2: Thank you very much. The comments and suggestions made by you have helped us improve the manuscript significantly. Information about vaccines has been added in Methodology 2.2. The change has been highlighted in yellow in the revised manuscript.
Point 3: Author need to provide information regarding the blood withdraw from the mice.
Response 3: Thank you very much. The comments and suggestions made by you have helped us improve the manuscript significantly. Information about vaccines has been added in Methodology 2.1. The change has been highlighted in yellow in the revised manuscript.
Point 4: Author need to explain the basis of administration of vaccine for four times over 4 weeks in the current study.
Response 4: Thank you very much. We had performed a total of two-dose immunization, given every two weeks. Previous studies had shown that one dose is less effective, so we carried out a two-dose immunization strategy.
Point 5: At first Short signs should be elaborate and utilize the entire Manuscript Draft. Example nCoV and at 33 line the same error was observed.
Response 5: Thanks for pointing out our careless mistakes. All changed abbreviations throughout the manuscript have been highlighted in blue.
Point 6: Please check 47 line, it has sentence error
Response 6: Thanks for pointing out our careless mistakes. We have modified “canparasitize” to “can parasitize” on line 48 based on your suggestion.
Point 7: Further Sampling /Number of mice has been used in the present study should be mentioned and Why sampling was different from figure 3 to Figure 4.
Response 7: Thanks for pointing out our careless mistakes. Due to our negligence, two sets of data were missed when processing the data, and now it has been supplemented in Figure 3.
Point 8: After how many days after serum was collected and tested for further follow up studies should mention in Methodology.
Response 8: Thank you very much. The comments and suggestions made by you have helped us improve the manuscript significantly. Information about vaccines has been added in Methodology 2.1. The change has been highlighted in yellow in the revised manuscript.
Point 9: For the betterment of this manuscript author should include the depicted time scale of immunization and blood withdraw should be included.
Response 9: Thank you very much. The comments and suggestions made by you have helped us improve the manuscript significantly. Information about vaccines has been added in Methodology 2.1. The change has been highlighted in yellow in the revised manuscript.
Point 10: In figure 2 Scale bar or Magnification of image was missing.
Response 10: Thanks for pointing out our careless mistakes. We have added to the Figure 2.
Point 11: In the Methods the source of Cell lines and Passage should be clarified. Author has plated 5 *105 cells per ml but at the same time in the methods mention author mentioned good condition. Could you please elaborate good condition?
Response 11: Thank you very much. The comments and suggestions made by you have helped us improve the manuscript significantly. "In good condition" specifically refers to the cell health and good cell vitality, so we changed it to "In good health" in Line 115.
Point 12: Author should describe after how many hours inoculate was treated should be explained in the methodology
Response 12: Thank you very much. The comments and suggestions made by you have helped us improve the manuscript significantly. Information about vaccines has been added in Methodology 2.1. The change has been highlighted in yellow in the revised manuscript.
Point 13: In the methodology adeno virus titer determination, author should mention the microscopic field chosen the calculation.
Response 13: Thank you very much. The comments and suggestions made by you have helped us improve the manuscript significantly. We have supplemented the microscope information in line 123 and line 129.
Point 14: In results section Figure 3 should categorized into A-J for simplifying viewer’s View.
Response 14: Thanks for pointing out our careless mistakes.We have added to the Figure 3.
Point 15: The author need to recheck check IL beta and /KC/GRO calculations. Since the down ward is error bar is too high is it falls under 0.005 and 0.023 statistics?
Response 15: Thank you for pointing out our careless mistakes. Due to our negligence, two sets of data were omitted when processing the cytokine data, which may lead to large errors. Now the supplementary data has been completed, and the t-test analysis is carried out with spss software, as shown in Figure 3.
Point 16: In the figure legends the author needs to mention how many animals were used per group should be mentioned. In the Figure 4 C why only three animals were used?
Response 16: Thank you very much. The comments and suggestions made by you have helped us improve the manuscript significantly. Live virus neutralizing antibody test data < 20 is considered negative, and data statistics will not be performed. Only three sets of positive data were measured in this study, so there are only three points.
Point 17: In the line 290, Reference need to include for his statement.
Response 17: Thank you very much. The comments and suggestions made by you have helped us improve the manuscript significantly. We have added the reference in Line 330.
Point 18: In the conclusion the author stated the adenovirus vaccine demonstrated that the vaccine has enough safety and effectiveness, but we did not see any type of the safety data in mice through out the study. Please address the above statement.
Response 18: Thank you very much. The comments and suggestions made by you have helped us improve the manuscript significantly. We use the human adenovirus type 5 vector, the safety of which has been extensively verified, and related products (Ad-p53) have been approved by sFDA.
Point 19: Why the first four reference style is different from the rest?
Response 19: Thanks for pointing out our careless mistakes. We have revised the first four reference style.
Reviewer 2 Report
This manuscript could be potentially interesting, however it is far from complete, and numerous gaps should be filled.
1. “In the presence of S1 protein, lung cells expressing human leukocyte antigen 42 (HLA)-E inhibited natural killer cell activation and IFN-γ _s_e_c_r_e_t_i_o_n [6]. Therefore, the 43 development of nCoV vaccines targeting the S protein S1 domain will increasingly 44 become one of the main preventive treatment strategies [7]”.
The references “6-7” are not sufficient to support the probable efficacy of a vaccine based on spike protein S1.
In addition, it would have been more suggestive, before proceeding to biotechnological process, carry out a bioinformatic analysis in order to assess both the degree of interaction with the specific ligand and any domain variations present in the current known variant strains.
Evaluation of the immunological response directed against adenovirus vector represents another important issue to be evaluated and discussed.
2. The data concerning the immunization protocols of the tested mice (number of animals, sex, age, etc.) are missing.
3. Immunofluorescence assay: negative controls, so as to exclude any phenomena of autofluorescence or anti-adenoviral vector reactivity, are missing.
4. The neutralization assay is not explained in detail so as to make it reproducible
There are, in addition, several typing errors.
Author Response
Response to Reviewer 2 Comments
This manuscript could be potentially interesting, however it is far from complete, and numerous gaps should be filled.
Point 1: “In the presence of S1 protein, lung cells expressing human leukocyte antigen 42 (HLA)-E inhibited natural killer cell activation and IFN-γ _s_e_c_r_e_t_i_o_n [6]. Therefore, the 43 development of nCoV vaccines targeting the S protein S1 domain will increasingly 44 become one of the main preventive treatment strategies [7]”.
The references “6-7” are not sufficient to support the probable efficacy of a vaccine based on spike protein S1.
In addition, it would have been more suggestive, before proceeding to biotechnological process, carry out a bioinformatic analysis in order to assess both the degree of interaction with the specific ligand and any domain variations present in the current known variant strains.
Evaluation of the immunological response directed against adenovirus vector represents another important issue to be evaluated and discussed.
Response 1: Thank you very much. The comments and suggestions made by you have helped us improve the manuscript significantly. We have made modifications in line 42-46. The human body will produce antibodies to the adenovirus vector used in our study, and some people have pre-existing antibodies of adenovirus in their bodies. The dose of the vaccine we tested is sufficient to overcome the pre-existing antibodies.
Point 2: The data concerning the immunization protocols of the tested mice (number of animals, sex, age, etc.) are missing.
Response 2: Thank you very much. The comments and suggestions made by you have helped us improve the manuscript significantly. Information about this vaccine has been added in Methodology 2.1. The change has been highlighted in yellow in the revised manuscript.
Point 3: Immunofluorescence assay: negative controls, so as to exclude any phenomena of autofluorescence or anti-adenoviral vector reactivity, are missing.
Response 3: Thanks for pointing out our careless mistakes. We have supplemented in Figure 2.
Point 4: The neutralization assay is not explained in detail so as to make it reproducible.
Response 4: Thank you very much. The comments and suggestions made by you have helped us improve the manuscript significantly. Details are supplemented in Methodology 2.9.1. The changes have been highlighted in yellow in the revised manuscript.
Reviewer 3 Report
The article submitted by Chen and colleagues describes the generation of an adenovirus-based vaccine against SARS-CoV-2. A truncated version of SARS-CoV-2 S1 has been introduced into the adenoviral backbone. Sera of immunized mice have been analyzed for neutralization capacities and humoral immune response. Although the development of novel and efficient vaccines is of high interest, this study has several limitations and flaws and is therefore not acceptable for publication in its current form.
The following points have to be addressed:
1. One major flaw of this manuscript is the limited information given on experimental procedures. The material and methods part lacks important information:
- There is no information given on the immunization of mice (e.g. breed, time schedule of immunization, dose etc). Such information is mandatory for immunization studies involving animal experiments.
- Line 77-78: “the sequence enconding SARS-CoV-2 amino acid 2-688”. Do you mean amino acids of SARS-CoV-2 S1?
- Line 90: What do you mean with “good condition”?
- Line 98: Which antibodies have been used?
- Line 109: What was the target of RT-PCR?
2. There are already some published studies focusing on adenovirus-based vaccines against SARS-CoV-2 that should be mentioned in the introduction and discussion. Especially the differences/ advantages between the vaccine described in this manuscript and the already published studies has to be discussed.
3. It has already been shown by others that adenovirus-based vaccines can be delivered orally or via aerosol inhalation and are capable of inducing mucosal antibody response. What are your reasons to stick to the i.m. admistration as especially adenovirus vaccines seem to be suitable candidates for oral/nasal administration?
4. The authors have only tested one particular S1 sequence. For other vaccines it has already been shown that the emergence of novel variants might diminish antiviral activity of neutralizing antibodies. Therefore, it is required to test more than one S1 sequence. Also, it is not clear which S1 sequence has been used for the generation of the vaccine.
5. Fig. 2A: Are the lines 1-6 different viral clones?
6. Fig. 2C: According to the picture, it seems that 100% of cell are infected. Is this true or is most of the staining background? Pictures of stained, uninfected cells are helpful for comparison.
7. Line 240-243: There is a discrepancy for IL-6. The corresponding graph shown in Fig. 3 does not show any significant decrease.
8. Line 243: The text refers to Fig. 3.
9. Fig. 4C: Did the group S1 14 days only include 3 animals or do all animals have the same values? If the values are identical, please provide an explanation why there is no variation at all.
10. Line 259: I have not read anything about “VSV-immunized mice” on the previous pages. Please clarify.
11. Line 329: The authors conclude that the vaccine had high efficacy in animal models. This hypothesis is not supported by functional data as infection studies in mice or other animals are missing.
Author Response
Response to Reviewer 3 Comments
The article submitted by Chen and colleagues describes the generation of an adenovirus-based vaccine against SARS-CoV-2. A truncated version of SARS-CoV-2 S1 has been introduced into the adenoviral backbone. Sera of immunized mice have been analyzed for neutralization capacities and humoral immune response. Although the development of novel and efficient vaccines is of high interest, this study has several limitations and flaws and is therefore not acceptable for publication in its current form.
The following points have to be addressed:
Point 1: One major flaw of this manuscript is the limited information given on experimental procedures. The material and methods part lacks important information:
Point 1.1: There is no information given on the immunization of mice (e.g. breed, time schedule of immunization, dose etc). Such information is mandatory for immunization studies involving animal experiments.
Response 1.1: Thank you very much. The comments and suggestions made by you have helped us improve the manuscript significantly. Information about vaccines has been added in Methodology 2.1. The change has been highlighted in yellow in the revised manuscript.
Point 1.2: Line 77-78: “the sequence enconding SARS-CoV-2 amino acid 2-688”. Do you mean amino acids of SARS-CoV-2 S1?
Response 1.2: Thank you very much. The comments and suggestions made by you have helped us improve the manuscript significantly. The sentence “the sequence enconding SARS-CoV-2 S1 amino acid 2-688” has modified to “the sequence enconding SARS-CoV-2 amino acid 2-688”.
Point 1.3: Line 90: What do you mean with “good condition”?
Response 1.3: Thank you very much. The comments and suggestions made by you have helped us improve the manuscript significantly. "In good condition" specifically refers to the cell health and good cell vitality, so we changed it to "In good health" in Line 115.
Point 1.4: Line 98: Which antibodies have been used?
Response 1.4: Thanks for pointing out our careless mistakes. We have made modifications. We have added the corresponding antibody information in line 123 and line 129.
Point 1.5: Line 109: What was the target of RT-PCR?
Response 1.5: Thanks for pointing out our careless mistakes. We have added the target in line 145.
Point 2: There are already some published studies focusing on adenovirus-based vaccines against SARS-CoV-2 that should be mentioned in the introduction and discussion. Especially the differences/advantages between the vaccine described in this manuscript and the already published studies has to be discussed.
Response 2: Thank you very much. The comments and suggestions made by you have helped us improve the manuscript significantly. We have added some related description in the Discussion section. The change has been highlighted in yellow in the revised manuscript.
Point 3: It has already been shown by others that adenovirus-based vaccines can be delivered orally or via aerosol inhalation and are capable of inducing mucosal antibody response. What are your reasons to stick to the i.m. admistration as especially adenovirus vaccines seem to be suitable candidates for oral/nasal administration?
Response 3: Thank you very much. The comments and suggestions made by you have helped us improve the manuscript significantly. This study is only an preliminary attempt, and We will continue to futher attempt the study of inhaled adenovirus vaccine. Vaccine projects based on the design of variant strains are also being carried out in our group.
Point 4: The authors have only tested one particular S1 sequence. For other vaccines it has already been shown that the emergence of novel variants might diminish antiviral activity of neutralizing antibodies. Therefore, it is required to test more than one S1 sequence. Also, it is not clear which S1 sequence has been used for the generation of the vaccine.
Response 4: Thank you very much. The comments and suggestions made by you have helped us improve the manuscript significantly. This study is only an preliminary attempt, and We will continue to futher attempt the study of inhaled adenovirus vaccine. Vaccine projects based on the design of variant strains are also being carried out in our group.
Point 5: Fig. 2A: Are the lines 1-6 different viral clones?
Response 5: Thanks for pointing out our careless mistakes. Lines 1-6 are repeated experiments, which we have descripted in line 257.
Point 6: Fig. 2C: According to the picture, it seems that 100% of cell are infected. Is this true or is most of the staining background? Pictures of stained, uninfected cells are helpful for comparison.
Response 6: Thank you very much. The comments and suggestions made by you have helped us improve the manuscript significantly. The MOI of our experiment is about 50, and the infection time has been more than 3 days, most of the cells are infected and obvious CPE could be observed, so it is not the background. Previous research on adenovirus was basically the same.
Point 7: Line 240-243: There is a discrepancy for IL-6. The corresponding graph shown in Fig. 3 does not show any significant decrease.
Response 7: Thank you for pointing out our careless mistakes. Due to our negligence, two sets of data were omitted when processing the cytokine data, which may lead to large errors. Now the supplement of data has been completed, and the t-test analysis is carried out with spss software, as shown in Figure 3. We also consider the change to be statistically significant.
Point 8: Line 243: The text refers to Fig. 3.
Response 8: Thanks for pointing out our careless mistakes. We change “Figure 4” to “Figure 3” in line 284.
Point 9: Fig. 4C: Did the group S1 14 days only include 3 animals or do all animals have the same values? If the values are identical, please provide an explanation why there is no variation at all.
Response 9: Thank you very much. The comments and suggestions made by you have helped us improve the manuscript significantly. Live virus neutralizing antibody test data < 20 is considered negative, and data statistics will not be performed. Only three sets of positive data were measured in this experiment, so there are only three points.
Point 10: Line 259: I have not read anything about “VSV-immunized mice” on the previous pages. Please clarify.
Response 10: Thanks for pointing out our careless mistakes. We changed “Analysis of neutralizing activities of serum from the VSV-immunized mice” to “Analysis of pseudovirus activities of serum from the Ad-S1-immunized mice” in line 299-300.
Point 11: Line 329: The authors conclude that the vaccine had high efficacy in animal models. This hypothesis is not supported by functional data as infection studies in mice or other animals are missing.
Response 11: Thank you very much. The comments and suggestions made by you have helped us improve the manuscript significantly. We only performed experiments in the mouse model, which has been modified for the “in mice model” in line 387. We will also conduct experimental research on other animal models in the future.
Round 2
Reviewer 1 Report
Thank you very much point to point addressing my concerns however following changes need to be applied in the current draft such
At the 81 and 82nd line (in the Methodology) blood can’t be collected from the eyeball it should be termed as a retro-orbital puncture.
In response 4 author addressed performing a total of two-dose immunization, given every two weeks. Previous studies had shown that one dose is less effective, so we carried out a two-dose immunization strategy., did you address or cited the previous study in the current manuscript Draft?
In response 9 author mentioned that he provided details in the methodology, but my suggestion is author for a better viewer experience these details should be depicted in figure 1.
· Please incorporate the recent study as a reference (Sheinin M, Jeong B, Paidi RK, Pahan K. Regression of Lung Cancer in Mice by Intranasal Administration of SARS-CoV-2 Spike S1. Cancers (Basel). 2022 Nov 17;14(22):5648. doi: 10.3390/cancers14225648. PMID: 36428739; PMCID: PMC9688283.) at 35 th line to strengthen the introduction.
Author Response
Response to Reviewer 1 Comments
Thank you very much point to point addressing my concerns however following changes need to be applied in the current draft such
Point 1: At the 81 and 82nd line (in the Methodology) blood can’t be collected from the eyeball it should be termed as a retro-orbital puncture.
Response 1: Thanks for pointing out our careless mistakes. We have made modifications at 80th and 81st line.
Point 2: In response 4 author addressed performing a total of two-dose immunization, given every two weeks. Previous studies had shown that one dose is less effective, so we carried out a two-dose immunization strategy., did you address or cited the previous study in the current manuscript Draft?
Response 2: Thank you very much. The comments and suggestions made by you have helped us improve the manuscript significantly. The corresponding references have been added at 78 th line.
Point 3: In response 9 author mentioned that he provided details in the methodology, but my suggestion is author for a better viewer experience these details should be depicted in figure 1.
Response 3: Thank you very much. The comments and suggestions made by you have helped us improve the manuscript significantly. We have supplemented Figure 1.
Point 4: Please incorporate the recent study as a reference (Sheinin M, Jeong B, Paidi RK, Pahan K. Regression of Lung Cancer in Mice by Intranasal Administration of SARS-CoV-2 Spike S1. Cancers (Basel). 2022 Nov 17;14(22):5648. doi: 10.3390/cancers14225648. PMID: 36428739; PMCID: PMC9688283.) at 35 th line to strengthen the introduction.
Response 4: Thank you very much for providing a good reference paper. The reference has been added at 35 th line.
Reviewer 2 Report
Accepted in the present form
Author Response
Thank you for your affirmation of our work and the revised manuscript .
Reviewer 3 Report
The authors have addressed all of my points and thereby increased the quality of their manuscript. Especially the methods are now described adequately. All questions and uncertainties have been addressed. I have no additional points to add and recommend the manuscript for publication.
Author Response

(The authors gave the same response as above.)
